# Factors associated with coronary heart disease in COPD patients and controls

**Christina D. Svendsen[1]\*, Karel K. J. Kuiper[2], Kristoffer Ostridge[3,4], Terje H. Larsen[2,5], Rune Nielsen[1,6], Vidar Hodneland[2], Eli Nordeide[1], Per S. Bakke[6], Tomas M. Eagan[1,6]**

**1** Department of Thoracic Medicine, Haukeland University Hospital, Bergen, Norway, **2** Department of Heart Disease, Haukeland University Hospital, Bergen, Norway, **3** Faculty of Medicine, Clinical and Experimental Sciences, University of Southampton, Southampton, United Kingdom, **4** Translational Science and Experimental Medicine, Research and Early Development, Respiratory & Immunology, BioPharmaceuticals R&D, AstraZeneca, Gothenburg, Sweden, **5** Department of Biomedicine, University of Bergen, Bergen, Norway, **6** Department of Clinical Science, University of Bergen, Bergen, Norway

\* chne@helse-bergen.no

## Abstract

### Background

COPD and coronary heart disease (CHD) frequently co-occur, yet which COPD phenotypes are most prone to CHD is poorly understood. The aim of this study was to see whether COPD patients did have a true higher risk for CHD than subjects without COPD, and to examine a range of potential factors associated with CHD in COPD patients and controls.

### Methods

347 COPD patients and 428 non-COPD controls, were invited for coronary computed tomography angiography (CCTA) and pulmonary CT. Arterial blood gas, bioelectrical impedance and lung function was measured, and a detailed medical history taken. The CCTA was evaluated for significant coronary stenosis and calcium score (CaSc), and emphysema defined as >10% of total area <-950 Hounsfield units.

### Results

12.6% of the COPD patients and 5.7% of the controls had coronary stenosis (p<0.01), whereas 55.9% of the COPD patients had a CaSc>100 compared to 31.6% of the controls (p<0.01). In a multivariable model adjusting for sex, age, body composition, pack-years, CRP, cholesterol/blood pressure lowering medication use and diabetes mellitus, the OR (95% CI) for having significant stenosis was 1.80 (0.86–3.78) in COPD patients compared with controls. In a similar model, the OR (95% CI) for having CaSc>100 was 1.68 (1.12–2.53) in COPD patients compared with controls. Examining the risk of significant stenosis and CaSc>100 among COPD patients, no variable was associated with significant stenosis, whereas male sex [OR 2.85 (1.56–5.21)], age [OR 3.74 (2.42–5.77)], statin use [OR 2.23 (1.23–4.50)] were associated with CaSc>100, after adjusting for body composition, pack-years, C-reactive protein, use of angiotensin converting enzyme (ACE) inhibitors or

**Data Availability Statement:** Data relevant for this study are available on Dryad at: https://doi.org/10.5061/dryad.2fqz612r4.

**Funding:** The author(s) received no specific funding for this work.

**Competing interests:** The authors have declared that no competing interests exist.

**Abbreviations:** COPD, chronic obstructive pulmonary disease; CHD, coronary heart disease; CCTA, coronary computed tomography angiography; CaSc, calcium score; ACE, angiotensin converting enzyme; ARBs, angiotensin receptor blockers; FVC, Forced vital capacity; $FEV_1$, forced expiration volume in 1 second; FMI, fat mass index; FFMI, fat free mass index; GFR, glomerular filtration rate; OR, odds ratio; SD, standard deviation.

angiotensin receptor blockers (ARBs), diabetes, emphysema score, GOLD category, exacerbation frequency, eosinophilia, and hypoxemia.

## Conclusion

COPD patients were more likely to have CHD, but neither emphysema score, lung function, exacerbation frequency, nor hypoxemia predicted presence of either coronary stenosis or CaSc>100.

## Background

Chronic obstructive pulmonary disease (COPD) is a leading cause of morbidity and mortality worldwide and includes phenotypic traits such as emphysema, exacerbation frequency, and respiratory failure. The rate of progression of COPD is heterogeneous, varying greatly between and within afflicted individuals. Many patients exhibit significant comorbidities, among them coronary heart disease (CHD), accompanied by both a high symptom burden and mortality [1,2]. Numerous previous studies have established that CHD is more common in COPD patients than in the general population [3–8]. Cardiovascular disease is the leading cause of death in COPD patients [9].

COPD and CHD share risk factors, especially cigarette smoking [10]. However, it is thought unlikely that the coexistence of COPD and CHD can be explained only by shared risk factors, since reduced lung function has been shown to be a risk factor for CHD independent of smoking [11]. Systemic inflammation has been demonstrated in several studies of COPD patients [12,13], and is known to be associated with atherosclerosis and plaque formation [14]. COPD patients with chronic hypoxemia and increased COPD exacerbation frequency may be especially prone to exhibit systemic inflammation and thus further increased risk for CHD. In addition, COPD patients with predominant emphysema have increased static hyperinflation, which may compromise cardiac function [13].

However, although these factors plausibly connect COPD to CHD, direct evidence that COPD development in itself leads to CHD is scarce. There may be at least two different explanations for this. First, COPD is a heterogenous disease, with large differences in disease manifestations. COPD disease severity is related to lung function decline, yet we know degree of emphysema can vary considerably between to patients with the same lung function. Thus, some patients may be more prone to develop CHD than others, depending on their manifestation of COPD. Second, previous studies have usually relied on self-report of presence of CHD, and not actual visualization of the coronary arteries. Thus, misclassification may also have confounded earlier studies.

In the current study from Western Norway, we performed a combined pulmonary CT scan and coronary CT angiography (CCTA) of our COPD patients and non-COPD controls, together with a thorough medical history, arterial blood gas and lung function measurements. The aim of this study was to see whether COPD patients did have a true higher risk for CHD than subjects without COPD, and to examine a range of potential factors associated with CHD in COPD patients and controls.

## Methods

### Study population and design

The participants were recruited as a cross-sectional sample from two previous patient-control cohorts: The MicroCOPD study and the follow-up phase of the GeneCOPD study. The

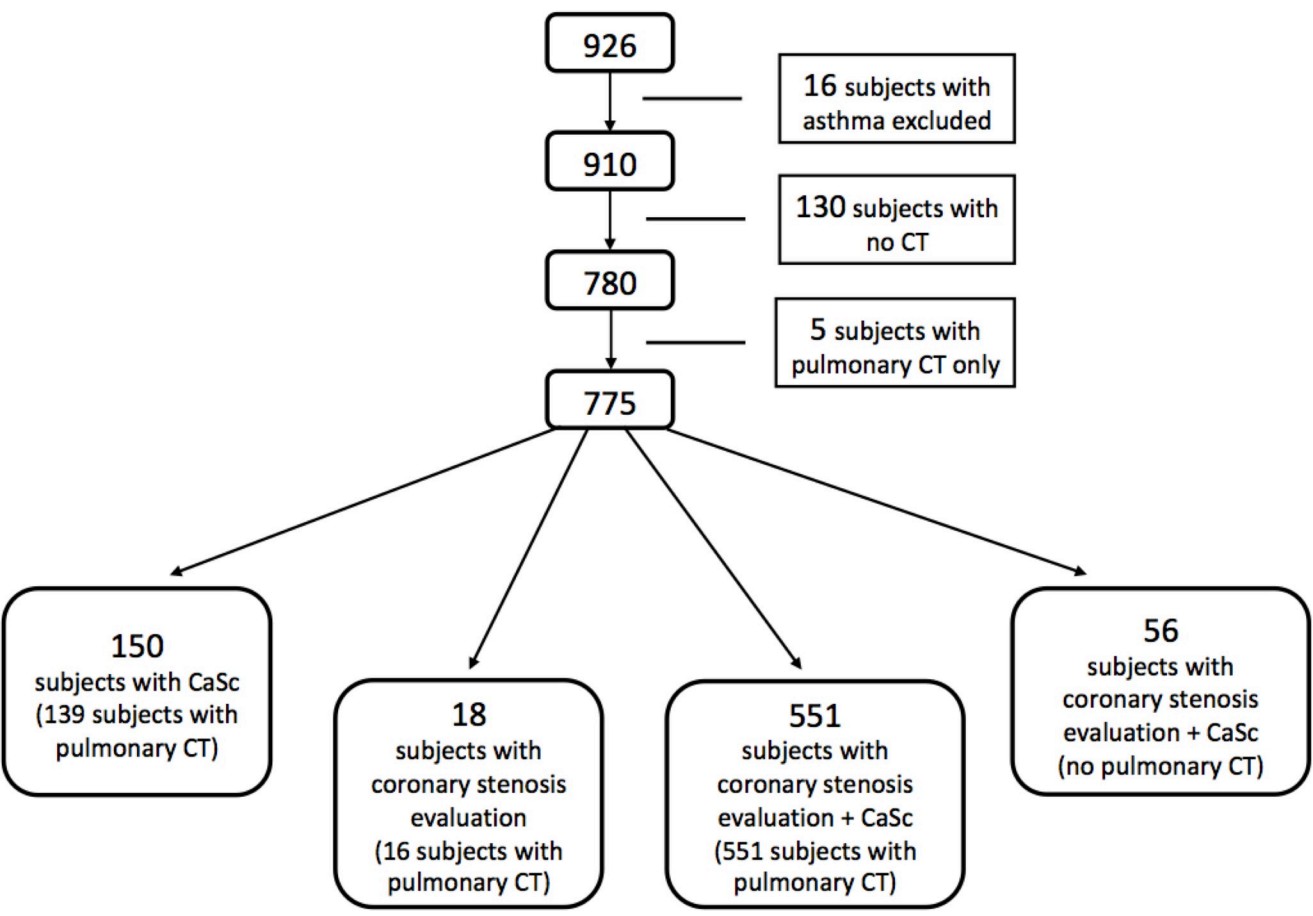

**Fig 1. Flow chart of study selection and final CT data availability.**

MicroCOPD Study was conducted 2012–2015 [15], and included 16 asthma patients whom were excluded from the current study, whereas the GeneCOPD follow-up 2013–2016 only included COPD patients and controls from the first GeneCOPD study in 2003–2004 [16].

In the original GeneCOPD study in 2003–2006, only half of the participants had pulmonary CT scans taken. At follow-up in 2013–206, more healthy controls than COPD patients were survivors, and due to cost restraints, not all controls without a baseline CT from 2003–2006 of the lungs were offered one at follow-up. There were 926 potential participants from the two cross-sectional samples, 16 asthma patients were excluded from the MicroCOPD study, 130 subjects had no CT taken, and for 5 subjects we only had a pulmonary CT, either due to allergy against contrast fluids, or finding calculated glomerular filtration rate (GFR) < 30 ml/min/ 1.73m$^2$. The final study sample included 775 participants, 347 COPD patients and 428 non-COPD controls (Fig 1).

The diagnosis of COPD was based on the Global Initiative for Chronic Obstructive Lung Disease (GOLD) guidelines [17] and required both spirometrically confirmed airflow obstruction, and a physician's assessment of the clinical history.

Subjects for whom the calcium score was above 500 or had concurrent coronary artifacts were prevented from assessment of coronary stenosis. In some instances, the export of single DICOM picture files from the IMPAX client was corrupted, and pulmonary CT scans were lost. Table 1 provides an overview of the study population, and numbers of available pulmonary CT measurements, coronary stenosis evaluations and calcium score measurements.

**Table 1. Overview of the study population who underwent pulmonary CT, CCTA, and measured CaSc.**

|  | COPD patients | Controls | Sum |
|---|---|---|---|
| Total number of participants | 347 | 428 | 775 |
| Participants with pulmonary CT | 315 | 391 | 706 |
| Participants with measured calcium scores | 333 | 424 | 757 |
| Participants with coronary stenosis evaluation (CCTA) | 254 | 371 | 625 |

Both studies were approved by the Norwegian ethical committee (GeneCOPD follow-up by REK-Vest, case number 2010/2015, MicroCOPD by REK-Nord, case number 2011/1307), and all participants gave written informed consent.

## Data collection

All patients and controls attended study visits at the outpatient clinic at Department of Thoracic Medicine, Haukeland University Hospital. All participants underwent lung function measurements on a Vitalograph 2160 Spirometer (Vitalograph Ltd, Maids Moreton, UK). Forced vital capacity (FVC) and forced expiration volume in 1 second ($FEV_1$) was measured after bronchodilation. Norwegian pre-bronchodilation reference values were used [18]. Arterial blood was taken and immediately analyzed on an ABL 720 radiometer for analyses of blood gases. Venous blood samples were taken, including hematology, cholesterols, and kidney function (creatinine for calculation of glomerular flow rate). In addition, body composition was assessed with bioelectrical impedance measurements using a Bodystat 1500 to calculate fat mass index (FMI) and fat free mass index (FFMI). Cachexia was defined as having a FFMI $< 14$ $kg/m^2$ in women and $< 17$ $kg/m^2$ in men which corresponds to the lower 95% CI in a normal population [19]. Similarly, obesity was defined as FMI $> 13.5$ $kg/m^2$ in women and $> 9.3$ $kg/m^2$ in men. Finally, a full medical history was obtained including respiratory symptoms, medication use, COPD exacerbation history in the last 12 months, and known comorbidities including diabetes.

## CT protocol

Both the coronary computed tomography angiography (CCTA) and pulmonary CT were performed with a 256-detector row Dual source Flash CT (Siemens®). Initially a scan of the heart region including coronary arteries was performed without contrast to determine calcium score, calculated using Agatstons coronary artery calcium score (CaSc) to describe density and extent of the calcification [20]. The CCTA was not performed if the CaSc was higher than 500 (n = 139, of which 85 were COPD patients). Among 561 subjects with non-significant stenosis, the median (IQR) CaSc was 15 (0–105), whereas among 46 subjects with significant stenosis the median CaSc was 243 (61–391), p-value = 0.0001, Kruskal Wallis test. As defined by Rumberger et al [21], a CaSc $> 100$ was used as an indicator of a high risk for coronary heart disease.

An interventional cardiologist and a cardiac radiologist evaluated the findings of coronary artery disease using the modified American Heart Association coronary segmentation model [22]. Lumen reduction was analyzed by measuring the diameter of the most stenotic part in the coronary artery. Confirmed coronary stenosis was defined as presence of stenosis (lumen reduction $> 50\%$).

The contrast medium Iomeron® 400 adjusted for body weight was used. Both the inspiratory and expiratory pulmonary scans were obtained with 0.5 mm intervals, and a reconstruction algorithm was added to both the CCTA and pulmonary CT scans. For classification of

emphysema on the pulmonary CT scans, 3D Slicer software [23] was used for density mask analysis, where significant emphysema was defined as > 10% of area below the density threshold -950 Hounsfield units (Hu), an accepted threshold validated with histopathology [24,25].

## Statistical analyses

Stata 14 was used for statistical analyses [26]. The Pearson $\chi^2$ test was used to compare categorical bivariate associations. Fisher exact test was used to compare variables with cell categories < 5. Multivariable logistic regression was performed to determine independent predictors of coronary stenosis and having a CaSc > 100. The analyses were first employed on the whole study group to compare COPD patients and non-COPD controls, adjusting for sex, age, body composition, smoking, c-reactive protein, use of statins and blood pressure lowering drugs, in addition to presence of diabetes. Afterwards, models were fitted to the COPD group only, to assess whether COPD disease characteristics could predict the presence of either coronary stenosis or a CaSc > 100. First-order interactions between sex, age, smoking and all other variables were tested for the COPD patients. Due to the large number of interactions (n = 66), a $p$-value of 0.01 was used, for all other analyses a $p$-value of < 0.05 was considered statistically significant.

## Results

Table 2 shows the study characteristics of the COPD and control subjects. Mean age among COPD patients and controls were 69.0 years (SD 7.9) and 64.8 years (SD 8.5) respectively. The COPD patients were older, less likely to have a normal body composition, had smoked more frequently and a larger load, and were more likely to use both cholesterol and blood pressure lowering drugs than control subjects. Among the COPD patients, 11.2% were GOLD category I, 53.5% GOLD category II, 25.9% GOLD category III, and 9.5% GOLD category IV. The COPD patients also had a higher calcium score and emphysema burden, in addition to a higher percentage afflicted with diabetes compared to the control subjects. Whereas 12.6% of COPD patients had confirmed coronary stenosis, only 5.7% of the controls did (p < 0.01). Considering CaSc, 55.9% of the COPD patients had a CaSc value >100, compared with 31.6% of the controls (p < 0.01).

Table 3A and 3B present potential predictors of presence of coronary stenosis or CaSc > 100 in COPD patients and controls respectively. Among a large range of potential predictors, male sex and age were associated with significant coronary stenosis > 50% for both groups (Table 3A). Further, prevalence of CaSc > 100 was associated with male sex, higher age, and statin use for both groups (Table 3B). GOLD category, emphysema score, and COPD exacerbation frequency were not associated with either coronary stenosis or an increased CaSc in the bivariate analyses.

The unadjusted and adjusted ORs with 95% CI for having coronary stenosis or a CaSc > 100 for COPD patients compared with controls is shown in Fig 2.

The controls had reference odds of 1, depicted by the stippled line. Adjustment was made for sex, age, body composition, pack years smoked, C-reactive protein, statin use, use of ACE inhibitors or ARBs, and diabetes. The COPD patients had an OR of 2.4 (1.35–4.27) for having coronary stenosis on the coronary CT scans compared with the controls, however after adjustment the association was reduced, and did not reach statistical significance (OR 1.80 (0.86–3.78, Fig 2 & Table 4A).

The COPD patients did have a statistically significant increased risk for having a CaSc >100, with an OR (95% CI) of 1.68 (1.12–2.53), Fig 2 & Table 4B. Male sex and higher age were associated with both coronary stenosis (Table 4A) and CaSc > 100 (Table 4B). CRP

**Table 2. Baseline characteristics of the study population in percentages.**

|  | COPD patients | Controls |  |
|---|---|---|---|
|  | (n = 347) | (n = 428) | p* |
| *Sex* |  |  | 0.89 |
| Women | 45.8 | 45.3 |  |
| Men | 54.2 | 54.7 |  |
| *Age categories (years)* |  |  | <0.01 |
| 40–59.9 | 12.7 | 34.1 |  |
| 60–69.9 | 43.8 | 42.5 |  |
| 70–90 | 43.5 | 23.4 |  |
| *Body composition* |  |  | <0.01 |
| Normal | 53.1 | 80.4 |  |
| Cachectic | 27.1 | 7.6 |  |
| Obese | 19.8 | 11.9 |  |
| *Smoking habits* |  |  | <0.01 |
| Never | 0.3 | 4.4 |  |
| Ex | 68.6 | 62.4 |  |
| Current | 31.3 | 33.2 |  |
| *Pack years per 10 years increase* |  |  | <0.01 |
| < 20 | 31.7 | 57.1 |  |
| 20–40 | 40.4 | 33.2 |  |
| 40+ | 28.0 | 9.8 |  |
| *Total cholesterol* (mmol/L) |  |  | 0.04 |
| 0–4.99 | 46.2 | 38.6 |  |
| 5–5.99 | 29.2 | 30.9 |  |
| 6–6.99 | 17.3 | 24.8 |  |
| 7 + | 7.2 | 5.6 |  |
| *Using statins* |  |  | <0.01 |
| Yes | 32.0 | 21.8 |  |
| *Using either ACE inhibitors or ARBs* |  |  | 0.35 |
| Yes | 16.4 | 14.0 |  |
| Having significant coronary stenosis |  |  | <0.01 |
| Yes | 12.6 | 5.7 |  |
| *Calcium score (HU)* |  |  | <0.01 |
| 0–100 | 44.1 | 68.4 |  |
| > 100 | 55.9 | 31.6 |  |
| *Emphysema as evaluated by CT (> 10% less than 950Hu)* |  |  | <0.01 |
| Yes | 22.9 | 0.26 |  |
| *C-reactive protein (mg/L)* |  |  | <0.01 |
| < 5 | 68.0 | 90.4 |  |
| ≧ 5 | 32.0 | 9.6 |  |
| *Eosinophilia (≥0.3*10^9 cells/L)* |  |  | <0.01 |
| Yes | 36.1 | 23.8 |  |
| *Diabetes* |  |  | <0.01 |
| Yes | 11.6 | 5.8 |  |

*Chi-square test for categorical variables.

**Table 3A. The prevalence of significant (>50% lumen reduction) coronary stenosis among 347 COPD patients and 428 non-COPD controls as evaluated from coronary CT scans.**

| | COPD patients | | | Controls | |
|---|---|---|---|---|---|
| | n (%) | p* | | n (%) | p* |
| *Sex* | | 0.04 | | | 0.02 |
| Women | 10 (8.1) | | | 5 (2.8) | |
| Men | 22 (16.8) | | | 16 (8.4) | |
| *Age categories (years)* | | 0.004† | | | 0.04 |
| 40–59.9 | 1 (2.4) | | | 5 (3.5) | |
| 60–69.9 | 12 (9.8) | | | 8 (4.9) | |
| 70–90 | 19 (21.4) | | | 8 (11.9) | |
| *Body composition* | | 0.37 | | | 0.33† |
| Normal | 14 (10.1) | | | 19 (6.4) | |
| Cachectic | 9 (15.5) | | | 1 (4.0) | |
| Obese | 8 (17.0) | | | 0 (0) | |
| *Smoking habits* | | 0.27† | | | 0.30† |
| Never | 0 (0) | | | 2 (13.3) | |
| Ex | 25 (14.8) | | | 13 (5.7) | |
| Current | 7 (8.3) | | | 6 (4.7) | |
| *Pack years* | | 0.35 | | | 0.07† |
| < 20 | 6 (8.3) | | | 11 (5.6) | |
| 20–40 | 14 (14.3) | | | 2 (1.8) | |
| 40+ | 11 (16.2) | | | 3 (11.1) | |
| *Total cholesterol (mmol/L)* | | 0.12† | | | 0.02† |
| 0–4.99 | 16 (14.2) | | | 15 (10.6) | |
| 5–5.99 | 4 (5.6) | | | 3 (2.6) | |
| 6–6.99 | 8 (16.0) | | | 3 (3.3) | |
| 7 + | 4 (21.1) | | | 0 (0) | |
| *Using statins* | | 0.49 | | | 0.01 |
| No | 21 (11.7) | | | 13 (4.3) | |
| Yes | 11 (14.9) | | | 8 (12.1) | |
| *Using either ACE inhibitors or ARBs* | | 0.21 | | | 0.18 |
| No | 25 (11.5) | | | 16 (5.0) | |
| Yes | 7 (18.9) | | | 5 (9.6) | |
| *C-reactive protein (mg/L)* | | 0.02 | | | 0.12† |
| < 5 | 16 (9.3) | | | 17 (5.1) | |
| ≧ 5 | 16 (19.8) | | | 4 (11.4) | |
| *Diabetes* | | 0.01 | | | 0.29† |
| No | 25 (10.9) | | | 19 (5.4) | |
| Yes | 7 (29.2) | | | 2 (10.5) | |
| *Emphysema as evaluated by CT* | | 0.79 | | | |
| No | 21 (11.6) | | | 20 (5.9) | |
| Yes | 6 (13.0) | | | 0 (0) | |
| *GOLD category* | | 0.85 | | | |
| I/II | 22 (12.9) | | | | |
| III/IV | 10 (12.1) | | | | |
| *> 1 exacerbations last year* | | 0.27† | | | |
| No | 30 (13.8) | | | | |
| Yes | 2 (5.7) | | | | |

*(Continued)*

**Table 3A.** (Continued)

| Eosinophilia (≥0.3*10^9 cells/L) | | 0.97 | | |
|---|---|---|---|---|
| No | 21 (12.7) | | | |
| Yes | 11 (12.5) | | | |
| Blood gases | | 0.38† | | |
| pO2 < 8 kPa | 2 (16.7) | | | |
| pO2 8–9 kPa | 3 (6.8) | | | |
| pO2 > 9 kPa | 27 (14.0) | | | |

* Chi-square test.

† If cells have less than 5 cases for a comparison, a Fisher's exact test was chosen.

**Table 3B The prevalence of calcium score >100 HU among 347 COPD patients and 428 non-COPD controls as evaluated from coronary CT scans.**

| | COPD patients | | Controls | |
|---|---|---|---|---|
| | n (%) | p* | n (%) | p* |
| Sex | | 0.002 | | <0.001 |
| Women | 75 (47.2) | | 36 (18.7) | |
| Men | 111 (63.8) | | 98 (42.4) | |
| Age categories (years) | | <0.001 | | <0.001 |
| 40–59.9 | 11 (25.0) | | 18 (12.4) | |
| 60–69.9 | 65 (44.2) | | 53 (29.4) | |
| 70–90 | 110 (77.5) | | 63 (63.6) | |
| Body composition | | 0.17 | | 0.01 |
| Normal | 83 (50.6) | | 93 (27.9) | |
| Cachectic | 51 (59.3) | | 17 (53.1) | |
| Obese | 41 (63.1) | | 20 (40.0) | |
| Smoking habits | | 0.13† | | 0.85 |
| Never | 0 (0) | | 6 (31.6) | |
| Ex | 135 (58.7) | | 86 (32.6) | |
| Daily | 51 (50.0) | | 42 (29.8) | |
| Pack years | | 0.44 | | 0.01 |
| < 20 | 57 (57.0) | | 67 (30.3) | |
| 20–40 | 63 (51.2) | | 36 (28.6) | |
| 40+ | 51 (59.8) | | 21 (55.3) | |
| Total cholesterol (mmol/L) | | 0.09 | | 0.01 |
| 0–4.99 | 96 (51.6) | | 67 (41.4) | |
| 5–5.99 | 50 (50.5) | | 35 (26.5) | |
| 6–6.99 | 27 (47.4) | | 26 (24.8) | |
| 7 + | 13 (52.0) | | 5 (20.8) | |
| Using statins | | <0.001 | | <0.001 |
| No | 112 (48.3) | | 85 (25.3) | |
| Yes | 74 (73.3) | | 49 (55.7) | |
| Using either ACE inhibitors or ARBs | | 0.01 | | 0.23 |
| No | 147 (52.9) | | 111 (30.5) | |
| Yes | 39 (70.9) | | 23 (38.3) | |
| C-reactive protein (mg/L) | | 0.51 | | 0.99 |
| < 5 | 124 (54.6) | | 121 (31.6) | |
| ≧ 5 | 62 (58.5) | | 13 (31.7) | |
| Diabetes | | 0.66 | | 0.09 |
| No | 164 (55.6) | | 123 (30.7) | |
| Yes | 22 (59.5) | | 11 (47.8) | |

(Continued)

**Table 3A.** (Continued)

| | | | | | |
|---|---|---|---|---|---|
| *Emphysema as evaluated by CT* | | 0.10 | | | 0.32† |
| No | 127 (54.5) | | 122 (31.6) | | |
| Yes | 46 (65.7) | | 1 (100) | | |
| *GOLD category* | | 0.11 | | | |
| I/II | 113 (52.6) | | | | |
| III/IV | 73 (61.9) | | | | |
| *> 1 exacerbations last year* | | 0.24 | | | |
| No | 161 (57.1) | | | | |
| Yes | 23 (47.9) | | | | |
| *Eosinophilia ($\geq 0.3^* 10^9$ cells/L)* | | 0.35 | | | |
| No | 116 (54.0) | | | | |
| Yes | 70 (59.3) | | | | |
| *Blood gases* | | <0.001† | | | |
| $p$O2 < 8 kPa | 19 (82.6) | | | | |
| $p$O2 8–9 kPa | 43 (68.3) | | | | |
| $p$O2 > 9 kPa | 120 (50.0) | | | | |

* Chi-square test.

† If cells have less than 5 cases for a comparison, a Fisher's exact test was chosen.

indicating systemic inflammation was predicative for significant coronary stenosis, but not CaSc > 100.

Looking at predictors among COPD patients, no variables were significantly associated with coronary stenosis (Table 5A). Considering the chance of finding CaSc > 100 in the COPD patients, male sex, higher age and statin use were associated at a statistically significant level (Table 5B). Neither of the COPD characteristics predicted the chance for having coronary stenosis or CaSc as judged by CCTA in these models, and neither of the first-order interactions tested were statistically significant (data not shown).

## Discussion

In this study of both COPD patients and non-COPD controls who underwent pulmonary CT imaging and CCTA, we confirmed the increased presence of coronary artery disease in COPD patients. After adjustment for potential confounders, COPD patients had significantly higher odds for having an increased calcium score. However, in the same multivariable analyses, age, male sex, and CRP ≥5 was significantly associated with having significant coronary stenosis, and age, male sex and statin use for increased calcium score. Among COPD patients only, neither of the COPD phenotypic traits GOLD category, presence and severity of respiratory failure, or CT evaluated emphysema, predicted the chance of finding stenosis on CCTA or CaSc > 100.

Although the coexistence of COPD and CHD has been established in several large studies of different study designs [4,5,27], there has been concern regarding misclassification, both due to COPD having many different manifestations, but also since establishing CHD with certainty requires somewhat invasive procedures. To date, few studies have performed coronary angiography in COPD patients. In a North American study of 351 COPD patients and 122 patients with interstitial lung disease (ILD) undergoing evaluation for lung transplantation, 60% had CHD of those in which coronary angiography was performed [6]. Interestingly, the prevalence of CHD was similar in COPD and ILD patients, although the smoking load was threefold higher among COPD patients.

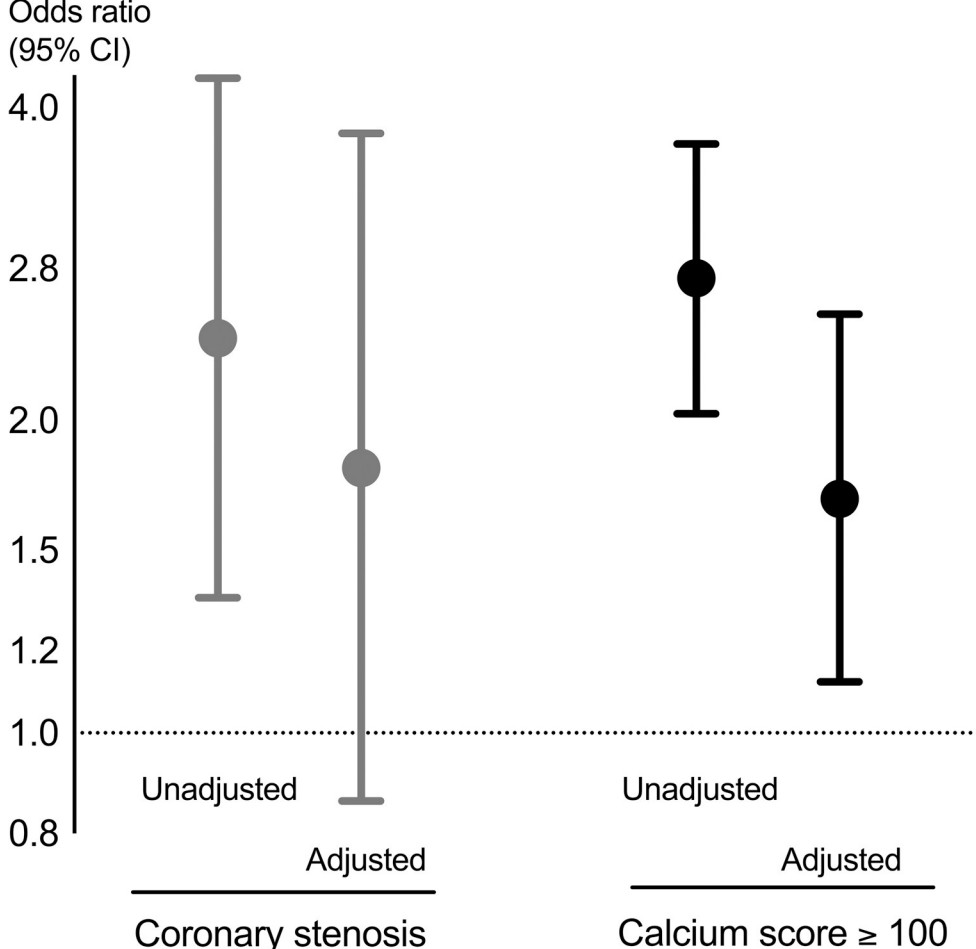

**Fig 2. The unadjusted and adjusted OR (95% CI) for having coronary stenosis and CaSc > 100 in COPD patients compared with non-COPD controls.**

In another study from Brazil, the prevalence of CHD was 88% among 101 COPD patients compared with 45% among 109 non-COPD controls [28]. In this study smoking habits varied greatly between study groups, and the prevalence of CHD in both groups were quite high. This may be due to a selection of patients under suspicion of heart disease, as is a likely cause of a high CHD prevalence in the largest study to date; a retrospective cohort study of 26,137 patients who underwent cardiac intervention and revascularization in Alberta, Canada [29]. In this registry study without spirometry data, the prevalence of CHD was 87% in COPD patients and 89% among non-COPD patients, however with more severe CHD among the COPD patients. In none of the three mentioned studies was emphysema evaluated with pulmonary CT.

Another sign of CHD is an increased coronary artery calcium score (CaSc) [30]. Just as with the few studies using coronary angiography, the few previous studies using CaSc have not yielded consistent findings. In a South-Korean study of 4905 men undergoing regular health measurements, lower FVC and $FEV_1$ was correlated with a higher CaSc [31]. However, in a general population sample from North America including both sexes and more comprehensive smoking characterization, neither $FEV_1$ nor emphysema were correlated with CaSc [32]. In an Italian lung cancer screening trial among 1,159 smokers, the lung function in the study

**Table 4A. The odds ratios (95% CI) for having significant coronary stenosis on CT angiography (n = 574).**

|  | OR | (95% CI) | p |
|---|---|---|---|
| *Study category* | | | |
| Controls | 1 | | |
| COPD | 1.80 | (0.86–3.78) | 0.12 |
| *Sex* | | | |
| Women | 1 | | |
| Men | 2.51 | (1.24–5.09) | 0.01 |
| *Age per 10 years increase* | 1.95 | (1.28–2.96) | 0.002 |
| *Body composition* | | | |
| Normal | 1 | | |
| Cachectic | 1.04 | (0.43–2.49) | 0.93 |
| Obese | 0.75 | (0.30–1.84) | 0.53 |
| *Pack years per 10 years increase* | 1.04 | (0.86–1.25) | 0.07 |
| *C-reactive protein (mg/L)* | | | |
| < 5 | 1 | | |
| ≧ 5 | 2.31 | (1.11–4.77) | 0.03 |
| *Using statins* | | | |
| No | 1 | | |
| Yes | 1.02 | (0.49–2.11) | 0.96 |
| *Using either ACE inhibitors or ARBs* | | | |
| No | 1 | | |
| Yes | 1.56 | (0.69–3.52) | 0.29 |
| *Diabetes* | | | |
| No | 1 | | |
| Yes | 1.97 | (0.78–4.95) | 0.15 |

**Table 4B The odds ratios (95% CI) for having calcium score > 100 Hu on CT angiography (n = 775).**

|  | OR | (95% CI) | p |
|---|---|---|---|
| *Study category* | | | |
| Controls | 1 | | |
| COPD | 1.68 | (1.12–2.53) | 0.014 |
| *Sex* | | | |
| Women | 1 | | |
| Men | 2.89 | (1.97–4.22) | <0.001 |
| *Age per 10 years increase* | 3.43 | (2.65–4.46) | <0.001 |
| *Body composition* | | | |
| Normal | 1 | | |
| Cachectic | 1.01 | (0.60–1.69) | 0.98 |
| Obese | 1.29 | (0.78–2.12) | 0.32 |
| *Pack years per 10 years increase* | 1.02 | (0.91–1.15) | 0.70 |
| *C-reactive protein (mg/L)* | | | |
| < 5 | 1 | | |
| ≧ 5 | 1.23 | (0.77–1.98) | 0.39 |
| *Using statins* | | | |
| No | 1 | | |
| Yes | 2.60 | (1.69–3.98) | <0.001 |
| *Using either ACE inhibitors or ARBs* | | | |
| No | 1 | | |
| Yes | 1.52 | (0.92–2.51) | 0.11 |

(*Continued*)

**Table 4A.** (Continued)

|  | **OR** | **(95% CI)** | **p** |
|---|---|---|---|
| *Diabetes* |  |  |  |
| No | 1 |  |  |
| Yes | 0.71 | (0.36–1.38) | 0.31 |

samples were lower than in the general population studies, however, no association between emphysema or $FEV_1$ and CaSc was found [33]. In this study non-gated low-dose CT with 5 mm slice thickness was used, which could have underestimated the CaSc.

Two studies have specifically examined COPD patients. In a matched case-control study of 81 COPD patients and 81 non-COPD controls from Switzerland, no difference in CaSc between groups were seen [34]. And in a sub-analysis from the ECLIPSE cohort, CaSc was found to be significantly higher among 672 COPD patients than 270 controls after adjustment for age, sex, ethnicity and pack-year smoking history [35]. Among COPD patients, neither $FEV_1$, GOLD category, nor exacerbation frequency was significantly related to CaSc. However, CaSc was calculated from pulmonary CT, not as part of a coronary CaSc evaluation [35].

The current study, with superior characterization of the presence of coronary heart disease [36], adds weight to the notion that COPD patients indeed have more CHD than non-COPD patients, and more than smoking history alone can explain. However, it appears that COPD severity in terms of lower $FEV_1$, presence of respiratory failure, or having a history of frequent exacerbations did not present an increased risk for having CHD, much the same as was seen in the ECLIPSE cohort. Neither did increased smoking in terms of pack-years present a clear risk for CHD or having a higher CRP, among COPD patients. However, CRP ≥5 was associated with having significant coronary stenosis, but not CaSc >100, among the study population. CRP is only an indirect measurement of systemic inflammation, yet the findings in this study was in concordance with the works of Jenny et al and Lin et al [37,38].

Use of cholesterol lowering drugs was significantly associated with CaSc >100 in our study. This is possibly because of primary or secondary prophylaxis treatment, but it is also worth considering increased CaSc after treatment with cholesterol lowering drugs such as statins as a representation of plaque repair and stabilization [39].

Thus, if COPD severity is not predictive of increased risk for CHD even though COPD patients as a group have more CHD, what could that mean? One possibility is a risk factor being at play early in the course of disease, linked to the start or development of COPD, rather than its progression. It is already known that smokers who quit still have a higher rate of decline in $FEV_1$ after smoking, once COPD is established, thus a similar phenomenon may be happening here. Which factors these may be are obviously unknown and not captured by the current study, but it is tempting to suggest factors of known relevance to CHD, and thus somehow linked to inflammation.

Second, it also means that clinicians need to keep in mind that having COPD, regardless of disease severity, carries with it an increased risk of CHD. This awareness is important to avoid oversight of symptoms like dyspnea and vague chest discomfort, which can easily be interpreted as symptoms caused by the known disease COPD.

The analyses in this study are associated with some limitations. First, causality cannot be established from a cross-sectional study. Second, study sample size may have been too low, and there is a chance for type II errors. Thus, we urge caution in interpreting all negative associations as proof of a non-causal relationship. Third, all of the subjects included in the study did not do a complete set of the CT scans for different reasons. 56 subjects were registered

**Table 5A. The odds ratios (95% CI) for having significant coronary stenosis on CT angiography among COPD patients (n = 205).**

| | OR | (95% CI) | p |
|---|---|---|---|
| *Sex* | | | |
| Women | 1 | | |
| Men | 2.72 | (0.97–7.56) | 0.06 |
| *Age per 10 years increase* | 1.74 | (0.94–3.23) | 0.08 |
| *Body composition* | | | |
| Normal | 1 | | |
| Cachectic | 2.71 | (0.79–9.34) | 0.11 |
| Obese | 1.92 | (0.63–5.84) | 0.25 |
| *Pack years per 10 years increase* | 0.97 | (0.75–1.26) | 0.82 |
| *Using statins* | | | |
| No | 1 | | |
| Yes | 0.60 | (0.21–1.71) | 0.34 |
| *Using either ACE inhibitors or ARBs* | | | |
| No | 1 | | |
| Yes | 2.64 | (0.80–8.71) | 0.11 |
| *C-reactive protein (mg/L)* | | | |
| < 5 | 1 | | |
| ≧ 5 | 2.19 | (0.80–5.97) | 0.13 |
| *Diabetes* | | | |
| No | 1 | | |
| Yes | 2.49 | (0.67–9.25) | 0.17 |
| *Percent area with < 950 Hu density (emphysemascore) per 1% increase* | | | |
| *GOLD category* | 1.02 | (0.95–1.08) | 0.62 |
| I/II | 1 | | |
| III/IV | 0.57 | (0.18–1.82) | 0.35 |
| *> 1 exacerbations the last year* | | | |
| No | 1 | | |
| Yes | 0.39 | (0.07–2.26) | 0.29 |
| *Eosinophilia (≥0.3* 10^9 cells/L)* | | | |
| No | 1 | | |
| Yes | 0.96 | (0.36–2.56) | 0.93 |
| *Respiratory failure (pO2 < 8 kPa)* | | | |
| No | 1 | | |
| Yes | 2.15 | (0.33–14.21) | 0.43 |

**Table 5B The odds ratios (95% CI) for having calcium score > 100 Hu on CT angiography among COPD patients (n = 271).**

| | OR | (95% CI) | p |
|---|---|---|---|
| *Sex* | | | |
| Women | 1 | | |
| Men | 2.79 | (1.51–5.14) | 0.001 |
| *Age per 10 years increase* | 3.69 | (2.39–5.70) | <0.001 |
| *Body composition* | | | |
| Normal | 1 | | |
| Cachectic | 0.70 | (0.32–1.49) | 0.35 |
| Obese | 1.35 | (0.64–2.85) | 0.43 |
| *Pack years per 10 years increase* | 0.96 | (0.81–1.13) | 0.60 |
| *Using statins* | | | |

*(Continued)*

**Table 5A.** (Continued)

| | | | |
|---|---|---|---|
| No | 1 | | |
| Yes | 2.42 | (1.27–4.63) | <0.01 |
| *Using either ACE inhibitors or ARBs* | | | |
| No | 1 | | |
| Yes | 2.29 | (1.01–5.19) | 0.047 |
| *C-reactive protein (mg/L)* | | | |
| < 5 | 1 | | |
| ≧ 5 | 1.05 | (0.55–1.99) | 0.89 |
| *Diabetes* | | | |
| No | 1 | | |
| Yes | 0.65 | (0.26–1.64) | 0.36 |
| *Percent area with < 950 Hu density (emphysemascore) per 1% increase* | 1.02 | (0.98–1.06) | 0.44 |
| *GOLD category* | | | |
| I/II | 1 | | |
| III/IV | 1.16 | (0.56–2.39) | 0.69 |
| *> 1 exacerbations the last year* | | | |
| No | 1 | | |
| Yes | 1.30 | (0.55–3.08) | 0.56 |
| *Eosinophilia (≥0.3\*10^9 cells/L)* | | | |
| No | 1 | | |
| Yes | 0.85 | (0.47–1.54) | 0.59 |
| *Respiratory failure (pO2 < 8 kPa)* | | | |
| No | 1 | | |
| Yes | 2.87 | (0.70–11.84) | 0.15 |

with no pulmonary CT because of a technical failure in transferring the images from the radiological database. However, this is unlikely to have affected the results because of the random nature of these subject's exclusion. Fourth, the study participants, mainly recruited from previous studies and from the thoracic outpatient clinic of Haukeland University Hospital, may have had a more severe disease burden than the COPD patients in the general population. However, all GOLD stages were represented. Fifth, CaSc is a surrogate marker for atherosclerosis in the coronary arteries, associated with increased risk for coronary artery stenosis and CHD. A CaSc of zero does not fully exclude the presence of significant coronary stenosis in patients with chest pain syndrome [40], a state that can be difficult to distinguish in a patient with COPD and CHD. CaSc also identifies coronary artery lesions containing calcium. However, identifying non-calcified lesions is somewhat limited [41]. It is possible that a CaSc of zero might underestimate the overall coronary lesion burden, but the majority of the patients included in our study underwent both CaSc and CCTA for coronary stenosis evaluation.

In conclusion, the current study confirms the added risk for CHD among COPD patients, but despite the extensive characterization of the study cohort, did not identify specific phenotypes at risk. Rather it points to the importance of having a high degree of suspicion for CHD even in COPD patients with mild lung function impairment.

## Acknowledgments

### Declarations

**Ethics approval and consent to participate**. Both the GeneCOPD and MicroCOPD studies were approved by the Norwegian ethical committee (GeneCOPD follow-up by REK-Vest, case

number 2010/2015, MicroCOPD by REK-Nord, case number 2011/1307). All participants received oral and written information prior to inclusion, and all participants gave written informed consent.

## Author Contributions

**Conceptualization:** Karel K. J. Kuiper, Kristoffer Ostridge, Terje H. Larsen, Per S. Bakke, Tomas M. Eagan.

**Data curation:** Kristoffer Ostridge, Vidar Hodneland, Tomas M. Eagan.

**Formal analysis:** Christina D. Svendsen, Tomas M. Eagan.

**Investigation:** Kristoffer Ostridge, Terje H. Larsen, Eli Nordeide.

**Methodology:** Karel K. J. Kuiper, Tomas M. Eagan.

**Project administration:** Eli Nordeide, Per S. Bakke, Tomas M. Eagan.

**Supervision:** Karel K. J. Kuiper, Kristoffer Ostridge, Rune Nielsen, Tomas M. Eagan.

**Writing – original draft:** Christina D. Svendsen.

**Writing – review & editing:** Karel K. J. Kuiper, Kristoffer Ostridge, Terje H. Larsen, Rune Nielsen, Vidar Hodneland, Eli Nordeide, Per S. Bakke, Tomas M. Eagan.

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
