## [Decision Letter · Decision Letter 0]

10 Jan 2022

PONE-D-21-35460The association between COPD phenotypes and coronary heart disease in an observational case control studyPLOS ONE

Dear Dr. Eagan,

Thank you for submitting your manuscript to PLOS ONE. After careful consideration, we feel that it has merit but does not fully meet PLOS ONE’s publication criteria as it currently stands. Therefore, we invite you to submit a revised version of the manuscript that addresses the points raised during the review process. As you can see, the reviewers have stated this is a valuble piece of work that merits to be published. But, as often, there are some revisions that has to be made. Please, respond to the reviewers' comments, point by point.

In addition, I will put special importance to;

1. Revise the aims

2. More clearly describe the selcetion of subjects (and the exclusions). For instance, if they have both asthma and COPD, are they then excluded??

3. A flow chart could bea possible addition.

4. Please, consider the power, when discussing non-significant results

We look forward to receiving your revised manuscript.

Med vennlig hilsen/ Best regards!

Kjell Torén, MD, PhD

Academic Editor

PLOS ONE

Journal Requirements:

Reviewers' comments:

Reviewer's Responses to Questions

**Comments to the Author**

1. Is the manuscript technically sound, and do the data support the conclusions?

Reviewer #1: Yes

Reviewer #2: Partly

2. Has the statistical analysis been performed appropriately and rigorously? 

Reviewer #1: Yes

Reviewer #2: Yes

3. Have the authors made all data underlying the findings in their manuscript fully available?

Reviewer #1: Yes

Reviewer #2: Yes

4. Is the manuscript presented in an intelligible fashion and written in standard English?

Reviewer #1: Yes

Reviewer #2: Yes

5. Review Comments to the Author

Reviewer #1: Svedsen and colleagues analyse the presence of coronary artery plaques and calcium score, respectively, in patients with COPD compared with controls without CPOD in a Norwegian cohort, with focus of analysis on potential risk factors for these outcomes among the patients with COPD. As outlined nicely in the paper, this topic is of interest given the large co-occurrence of cardiovascular disease and COPD and the relevance for morbidity and prognosis. Reading the paper, I had some comments and suggestions for strengthening the paper:

- A general comment is that the analysis in fact focuses on a range of potential predictors of the outcomes (stenosis and calcium score) - not only COPD phenotype. Suggest that is reflected in the title ('Factors associated with' or 'Predictors of ...' instead of COPD phenotype), especially as many of the findings not pertain closely to phenotype, rather factors more broadly. It also takes a while in Methods before the phenotype variables are introduced. Please revise (and clarify) the aim accordingly (now the aim is rather vague and differ in Abstract and the main text).

- Need to be clearer on the limitation in terms of sample size / power. Many well established risk factors for cardiovascular disease (CVD) did not "fall out" in the analysis -- however, this is not to be interpreted as these factors are not risk factors in COPD patients (COPD patients are "spared") - when, in fact, it is the other way around (not implying this is stated by the authors, it is not), but important that this limitation is made clear. Insufficient power likely to explain the absence of a lot of expected associations? - that also is a prolem for the relevance of the analysis as a whole, but the analysis is still valuable as it is well done and could be combined with other datasets for meta-analyses

- Factors in the models: please state how adjustment factors as well as factors to evaluate in the model were selected.

- Intro: states that COPD severity is graded based on lung function -- not sure that statement is entirely correct any more, please revise

- Aim: as mentioned above, make the aims similar and specific - the aim in the paper is not good - "... whether different COPD patients had an increased risk..." needs revision in terms of scientific contents and language (surely different patients will have different risks, it would be surprising if all COPD patients had the same risk, etc) - evaluate factors associated with... ?

- Methods: please state which patients that were included in the used databases - now it only says that 16 asthma patients were included (which was confusing at the first read); inclusion and exclusion criteria. Suggest to move the data on actually included number of participants to the start of Results.

- Why was the CaSc threshold set to 100 for the outcome?

- Tables: no need to include both the "yes" and the "no" categories for binary variables

- Methods and Results: how were the models specified to begin with (in terms of factors evaluated and adjusted for) and how were the final factors to report determined?

- How many had a SaSc >500 (and were excluded from CTA)? Potential influence on findings?

- Table 4: "for the chance of finding..." - please revise and specify

- One wonders how good CaSc is as outcome - how was the correlation within the outcomes (CaSc with stenosis)? Would be good with more data on that, including in Methods as rationale for the cut-off and use of that outcome

- Please revise the sentence "Since this is a cross-sectional study, we can only speculate", as well as the sentences with "which can easily be interpred as", and "makes it uncertain to establish"

Reviewer #2: The main purpose of this study was to investigate it if different COPD characteristics or phenotypes are associated with an increased risk of coronary heart disease. The conceptualization is really interesting and the overall planning of the study is sound and clear. I have no major concerns but a number of minor comments mainly about design and analyses.

1. Design and data collection

The design and data collection could be more clearly described. The title says case control study, but is it really? The study has two parts, the first investigating the prevalence of coronary atherosclerosis in patients with COPD and non-COPD controls and the second investigating the associations of specific COPD characteristics with the dependent variable of CHD. However, it is not like the groups are defined from the outcome CHD and compared about the risk factors and it is unclear how the controls were chosen. The authors refer to the original papers of the involved cohorts, but it is unclear from the present paper if matching of cases and controls have been performed. As for the second part, it is unclear if the independent variables are collected at the same time as the outcome variable or not. The paper would benefit a lot from a flow chart clarifying the design, timing, the contributions from different other cohorts, and the attritions.

2. Comparison of CHD in COPD and non-COPD

The first major conclusion is that COPD patients are more likely to have objective measures of CHD. This is not surprising as the COPD patients are older, have more pack years and more diabetes. Have you considered comparing the groups by propensity score matching?

3. Logistic regression

In the second part, logistic regression uses different COPD characteristics as independent variables.

How were the explanation variables chosen? A priori based on subject knowledge matter or based on unadjusted analyses?

I find the research question very relevant, but it would have been even more interesting with more COPD characteristics as markers for systemic inflammation or other phenoptypes. Was chronic bronchitis or eosinophilic COPD investigated as independent variables? Emphysema was defined as 10% of the lungs or not, what happens if the analyses are repeated using emphysema a continuous score as independent variable? And was both hypoxemia and hypercapnia analysed? ACO? Non-smoking COPD? COPD with early onset or exposure during childhood/premature birth?

It would have been of great interest if the analyses did not only adjust for different characteristics/phenotypes, but also included stratification and interaction analyses to see if there is an effect modification by phenotype, ie- if factors associated with CHD differs between different phenotypes.

4. Power analysis

The second major conclusion is that no specific COPD characteristics were independently associated with a higher risk for CHD. Could this be a type 2 error? Was there a power analysis performed to ensure a reasonable number of participants?

5. Attrition

Was there any attrition analysis?

6. PLOS authors have the option to publish the peer review history of their article (what does this mean?). If published, this will include your full peer review and any attached files.

Reviewer #1: No

Reviewer #2: No

---

## [Author Response · Author response to Decision Letter 0]

27 Feb 2022

We have uploaded a word document with our point - to point responses to the two reviewers.

---

## [Editor Report · Decision Letter 1]

7 Mar 2022

Factors associated with coronary heart disease in COPD patients and controls

PONE-D-21-35460R1

Dear Dr. Eagan,

We’re pleased to inform you that your manuscript has been judged scientifically suitable for publication and will be formally accepted for publication once it meets all outstanding technical requirements.

Kind regards,

Kjell Torén, MD, PhD

Academic Editor

PLOS ONE

Additional Editor Comments (optional):

Dear dr Eagan!

I think you and your co-authors satisfactory have addressed the comments by the reviewers. When you do not agree, you have argued quite convincingly for your sake.

Hence, the manuscript is accepted.

---

## [Editor Report · Acceptance letter]

29 Mar 2022

PONE-D-21-35460R1 

Factors associated with coronary heart disease in COPD patients and controls 

Dear Dr. Eagan:

I'm pleased to inform you that your manuscript has been deemed suitable for publication in PLOS ONE. Congratulations! Your manuscript is now with our production department. 

Kind regards, 

on behalf of

Dr. Kjell Torén 

Academic Editor

PLOS ONE